# Evidence for College Students’ Decreasing Sense of Belonging over Time: Direct and Moderated Results

**DOI:** 10.3390/ijerph22040472

**Published:** 2025-03-22

**Authors:** Joshua K. Brown, Lauren M. Papp

**Affiliations:** 1Department of Psychology, Southern Utah University, Cedar City, UT 84721, USA; 2Department of Human Development and Family Studies, University of Wisconsin-Madison, Madison, WI 53706, USA; papp@wisc.edu

**Keywords:** college students, belonging, longitudinal, campus life

## Abstract

College students’ sense of belonging with their institution has been established as an important factor that improves their health and well-being. However, the extent to which students’ sense of belonging changes over their college experience—if at all—needs further examination. The current study employs a longitudinal design utilizing data repeatedly collected from the same students (*N* = 355; male 33.0% and female 67.0%; White 83.2%; baseline age range: 18–21 years) to examine changes in their sense of belonging over time. We also examined the extent to which different participant characteristics (i.e., sex, race/ethnicity, first-generation student status, and affiliation with Greek life organizations) are associated with change in belonging over time. College students’ belonging directly decreased over time, and race/ethnicity moderated this change such that being a student from a minoritized racial/ethnic group was associated with steeper drops in belonging. Educators and researchers are encouraged to further research this topic to determine how their classrooms, departments, and institutions can help foster students’ belonging and maintain that belonging over time.

## 1. Introduction

Students’ academic performance, health, and well-being are primary concerns for educators, administrators, and policymakers across all levels of influence. College students are an especially important student population [1] due to the crucial developmental period when many of them attend college (i.e., exiting adolescence into young adulthood [2]) and the robust effects that choices and behaviors during this period can have on later outcomes (e.g., careers, relationships, injuries, chronic illnesses; see [3,4,5,6,7]). Emerging adulthood can be rife with challenges and previous research found that many college students in this developmental period report distress [2,8,9,10], physical illnesses [11], and mental health problems, especially in relation to the COVID-19 pandemic [12,13,14]. Despite these issues, an important factor can help buffer against unfortunate outcomes: college students’ sense of belonging [15] with their college or university student group [3,16,17,18,19,20]. Strayhorn [20] further defines sense of belonging cognitively as a sense of closeness with a community that is heightened for students by perceived social support from peers and teachers, and its absence as “a sense of alienation, rejection, social isolation, or marginality” (p. 2). As reported in a nationally representative survey of first-year U.S. college students, a strong sense of belonging motivated students to access campus resources, enhanced social support, and led to better academic performance [21]. Using data collected from a larger study (see [22]), the current study is well situated to examine how college students’ sense of belonging changed across five consecutive time points evenly spaced six months apart and to determine whether these trajectories were affected by certain demographic factors and student life characteristics.

### 1.1. College Students’ Sense of Belonging

Given the developmental importance associated with students’ experiences during college, many factors that affect these experiences have been examined. An especially important construct that has emerged in the literature is college students’ sense of belonging [17,19,20,23,24] as a member of the university’s student population. Our operationalization comes from Strayhorn ([20] p. 4):

“In terms of college, sense of belonging refers to students’ perceived social support on campus, a feeling or sensation of connectedness, and the experience of mattering or feeling cared about, accepted, respected, valued by, and important to the campus community or others on campus such as faculty, staff, and peers.”

Students’ sense of belonging offers a useful model for understanding how elements that students bring into their academic experiences (e.g., race, gender identity, socioeconomic status) relate to the elements present within an academic institution (e.g., policies, pedagogies, resources) and, ultimately, to crucial outcomes (e.g., academic performance, career trajectory, health, and well-being).

Students’ sense of belonging is a primary predictor of student success, health, and well-being [3,14,17,19]. Multiple outcomes associated with college students’ belonging have been assessed: students who reported a stronger sense of belonging better persisted through college to graduation [17,19], found it easier to meet explicit and implicit academic expectations (e.g., joining research teams, engaging with classmates, and collaborating with faculty; [2,7,18]), and were buffered from academic stress [10,24,25]. Tholen and colleagues [24] also found that sense of belonging was significantly associated with better mental health. Furthermore, higher grade achievement was shown among young women in STEM fields with more belonging from a social identity intervention [18], among Black students who completed a first-year belongingness intervention [26], and among Chinese students reporting more belonging [27].

Recent research has shown that belonging was relatively high among college students [23]. Despite the importance of belonging as a construct, previous research has typically been limited to analyzing measurements collected according to one-time (e.g., [2,24]), cross-sectional (e.g., [4]), or temporally limited longitudinal designs (e.g., [28,29]). For example, Hausmann et al. [17] tested an intervention to enhance sense of belonging and found it amenable to changes within the first year of college among both Black and White college students, although the levels declined among the control and intervention groups over 9 months. This decline in belonging may progress beyond the first year of college. If so, then we would expect to see belonging decrease over students’ college careers in a longer longitudinal investigation.

Despite the important research reported, the field remains limited by the lack of longitudinal investigations; indeed, scholars have called for more studies using repeated measures covering longer periods of students’ lives to better understand the extent to which one’s sense of belonging levels change over time (see [4,11,14]). Some research has explored longitudinal patterns across two time periods: Nuñez [28] used structural equation modeling and showed that positive diversity experiences and engagement in the campus community were both associated with an increased sense of belonging among Latino college students from their first to second year of college; Pittman and Richmond [29] showed that students transitioning into college during their first year showed more positive benefits when they reported increase sense of belonging from time 1 to time 2. In a notable recent longitudinal study covering three time points, Ruedas-Gracia et al. [23] examined change in belonging among college students (*N* = 136) at a private university in the US. Their results showed belonging levels (assessed three times across 4 years) remained stable over time. However, despite being collected during the COVID-19 pandemic, a limitation was the lack of estimation of effects associated with the COVID-19 pandemic. Indeed, the researchers called for the need for additional research to specifically examine COVID-19’s effect on belonging [23]. Also, in their sample, the first wave of data collection did not coincide with matriculation (i.e., when students start college) and the three measurement periods were unequally spaced, leaving important questions about linear trajectories of belonging across the early college years. Research spanning the pandemic’s onset along with college students’ progression would be well situated towards addressing gaps and examining the full breadth of changes–or lack thereof—in belonging among college students. Our study expands on the limited periods of longitudinal investigation from the previous research by utilizing five equally spaced time points covering the breadth of students’ collegiate tenure.

### 1.2. Student Characteristics Affecting Sense of Belonging

College students’ sense of belonging does not exist in a vacuum and can be affected by many factors associated with their characteristics and experiences in college. Background demographic factors have substantial associations with belongingness [20,23]. Gopalan and Brady [21] identified that women and people with minoritized racial/ethnic identities reported lower belonging on 4-year college campuses in a nationally representative sample of first-year college students. On White-majority and historically male-majority college campuses though, students from minoritized groups may experience greater threats to their sense of belonging than students from historical majority groups, thus reducing their sense of belonging on campus [18,30,31,32,33].

The larger study that we drew from [22] also included other factors that may affect college students’ sense of belonging, including their first-generation college student status and their affiliation with Greek life organizations. First-generation college students report having trouble fitting in on college campuses [21]. This contributes to academic worry and leads to problems with retention for first-generation college students [34]. Accordingly, first-generation status may negatively affect college students’ sense of belonging such that these students show a decrease in belonging over time. In contrast, affiliation with Greek life organizations may positively affect college students’ belonging due to the camaraderie and frequent association with other college students simultaneously affiliated with fraternities or sororities, dependent on organizational involvement [35]. This varies over time though, given that students may enter and exit Greek life organizations over their college career. Frequent social events, intramural associations, and continual networking may thus act as a “belongingness boon” for students such that they show increases in belonging when they are affiliated with Greek life organizations [36].

### 1.3. Current Study

The current study investigated the extent to which students’ sense of belonging changes over time and whether these direct changes might be moderated by student characteristics (i.e., participant sex, race/ethnicity, first-generation status, and affiliation with Greek-life organizations). We employ a longitudinal design utilizing data repeatedly collected over five waves across multiple years from students at a large public university in the Midwestern US (*N* = 355; see [22]). Our design allowed us to account for potential effects of time in college and the COVID-19 pandemic on belong levels, thus improving prior work. The current study addresses previously identified gaps by examining changes in students’ sense of belonging levels across a large sample longitudinally surveyed over 3+ years in models that specifically account for time since college enrollment and any effect of assessment during COVID-19.

Our study takes advantage of data collected for a different purpose (see [22]). The original study was funded by the National Institute on Drug Abuse of the National Institutes of Health [22]. Secondary analyses of existing data are relatively common [13,16,37] and offer excellent opportunities to test further theories and hypotheses [38].

In terms of current hypotheses, in the direct model across all participants, we predicted that college students’ sense of belonging would remain stable over time, given findings from Ruedas-Garcia et al. [23]. In addition, a moderation model tested change over time for hypothesized participant characteristics; e.g., expected differences such that belonging levels would increase over time among those reporting Greek-life affiliation and decrease among women (compared to men), those from minoritized racial/ethnic groups, and those reporting first-generation student status.

## 2. Materials and Methods

### 2.1. Participants and Data Collection Procedure

This study was drawn from a larger longitudinal study about factors that affect college students’ health and well-being (Papp, PI; [22]); the OSF project registration for that study includes additional approach information and sample size justification. The university’s Institutional Review Board approved this project, and we obtained a NIH certificate of confidentiality. The Midwestern research university campus from which the sample was drawn has more than 30,000 undergraduates and around 10,000 graduate students, with slightly more women than men enrolled as undergraduates. College students ages 18–21 were recruited in their first year or second year of enrollment through ongoing recruitment efforts via advertisements on campus and the surrounding community. None of the participants completed surveys at the same time. Instead, participants completed the first survey after their individual recruitment and screening then each subsequent follow-up survey was sent to participants via email every 6 months. Participants completed informed consent procedures at the start of their Time 1 (T1) laboratory session. Participants were paid for completing each survey. At T1, participants reported the following racial/ethnic identities: White: 83.2%, Asian: 5.7%, Black: <2% (exact percentage not reported due to low frequency), American Indian or Alaska Native: <2%, Selected more than one race: 7.1%, Selected “Other”: <2%; Latinx/Hispanic Background: 6.7%. See Table 1 for further demographic information. Our sample’s demographic characteristics represent our university’s demographic make-up.

On the T1 assessment, surveys assessed multiple outcome and predictor domains (see Section 2.2); *N* = 355 participants completed the T1 survey [22]. After completing the T1 survey, participants were compensated USD 250. The same outcomes were assessed via 4 online follow-up surveys administered every 6 months for 2 years between Fall 2017 and Summer 2022 (T2: 99%, 353/355; T3: 98%, 348/355; T4: 98%, 349/355; T5: 97%, 346/355). Participants were compensated after each subsequent survey was completed: USD 90 for T2, USD 100 for T3, USD 110 for T4, and USD 120 for T5. Our individual payments were approved by our Institutional Review Board. Furthermore, our payment aligns with ethical considerations of social beneficence, as elucidated by Różyńska [39]. Analyses are based on responses over T1–T5. We have 355 participants across 5 waves with 1757 total data points.

### 2.2. Measures and Coding

#### 2.2.1. Time

Each survey (T1–T5) recorded participants’ survey completion date. We computed a person-specific time variable that centers each participants’ matriculation (i.e., first semester of college) as TIME = 0 and each surveys’ responses (T1–T5) are coded to represent the number of months elapsed from matriculation (TIME = 0 + X; Range: 0–37 months). Students matriculating in Fall semesters were coded as starting in September of that given year (e.g., September 2018), while those matriculating in Spring or Summer semesters were coded as starting in January or May of that given year, respectively (e.g., January 2018 or May 2018).

#### 2.2.2. Sense of Belonging

In each survey (T1–T5), participants reported their sense of belonging as a student in relation to the campus community [30]. Participants were asked 5 questions that correspond to their sense of belongingness with the campus community on a 5-point Likert scale anchored with ‘Strongly Disagree’ to ‘Strongly Agree’. Example item: “I feel that I am a member of the campus community”. Items’ scores were summed, with higher scores representing higher sense of belonging ([30]; see Table 2 for descriptive statistics across waves T1–T5).

#### 2.2.3. Covariate: Study Enrollment

The original study oversampled participants who had endorsed any recent prescription drug misuse over the last three months during the recruitment screener as part of larger study goals (see [22]). Specifically, 300 (84.5%) participants were recruited who endorsed recent prescription drug misuse at screening; 55 (15.5%) participants who did not endorse recent prescription drug misuse were also recruited as a comparison group (see [22]). While we do not expect reliable links between recent prescription drug misuse and students’ sense of belonging, including this variable acknowledges that the sample was collected for a different purpose (see Section 1.3). As such, we decided to control for this variable due to the unique sample generated from the larger study. Previous research using this same sample has used the identical covariate [13,16]. Thus, statistical models accounted for the broader study’s sampling strategy.

#### 2.2.4. Covariate: COVID-19

To account for COVID-19 effects (see [16]), we created a dichotomous variable centered on the COVID-19 pandemic in our geographic region (12 March 2020). Months preceding COVID-19 were coded = 0 and months succeeding COVID-19 onset were coded = 1, and this variable along with its influence on the slope of belonging’s change were included as level-1 covariates.

#### 2.2.5. Predictors

Greek affiliation. Affiliation with Greek life organizations (i.e., sororities and fraternities) was reported by all participants across surveys and coded dichotomously. This variable varied across time, as some participants joined or left a fraternity/sorority across the five waves (T1: 31.8% Greek; T2: 30.7% Greek; T3: 29.9% Greek; T4: 26.2% Greek; T5: 21.4% Greek).

Participant sex. Self-reported participant sex at T1 was included as a person-level moderator. All participants reported either male (33.0%) or female (67.0%) as their sex, though other options were available for selection.

Race/ethnicity. Self-reported race/ethnicity at T1 was included as a person-level moderator. To provide adequate groups for comparison in analyses, we followed Carter et al. [40] and Barringer et al. [16] and combined all participants who reported racial-ethnic minority identities and recoded race/ethnicity as a dichotomous effect-coded variable: White, non-Hispanic student = −0.5 (79.6%), and racial-ethnic minority student = 0.5 (20.4%). This aggregation was necessary for data analysis but does mean that we regretfully cannot identify the varied experiences of students from different ethnic and racial groups. Indeed, previous research shows a wide variety of experiences among students who socially identified as Latino, Black, and/or Asian [28,31,32,41].

First-generation college status. Participants reported their parents’ educational attainment across T2–T5 surveys. Participants reported educational attainment for parent 1 and parent 2, though some participants only reported educational attainment for one parent. Participants were coded as non-first-generation status (=0; 85.4%) if one or both parents earned at least a 4-year college degree while participants were coded as first-generation status (=1; 14.0%) if neither parent earned at least a 4-year college degree. Our definition adheres to our institution’s definition for first-generation students. We examined first-generation student status across T2–T5 waves and determined that status did not change for any participant across those waves; thus, first-generation student status was considered a person-level variable.

### 2.3. Analysis Plan

Given the longitudinal nature of our data, we used multilevel growth modeling to examine the changes in our outcome, belonging, across all levels over time, nesting repeated measurements (level-1) within participants (level-2) [42,43,44]. Along with each level’s unique effects, multilevel growth models provide important information about hypothesized cross-level effects (such as our hypothesized cross-level moderations of change over time by demographic characteristics) that cannot be easily accounted for using other analytic techniques. A 2-level model was used. Level-1 predictors include fixed and random parameters representing time since matriculation in months (TIME), time-varying Greek affiliation, and the COVID-19 onset indicator covariate. The second level accounts for between-person differences, including person-level sex, race/ethnicity, and first-generation student status along with the study enrollment covariate. Unstructured covariance structures and standard errors robust to violations of normality were used.

We used a bottom–up approach for building the multilevel growth model, following suggestions from Kristjansson et al. [44] and Burant [45]. First, we started with the null model and computed an intraclass correlation coefficient (ICC) to determine whether multilevel modeling was an appropriate analysis strategy. The ICC = .581, indicating that 58.1% of the variance in belonging was attributable to between-person differences, an appropriate amount for multilevel modeling [43]. Next, we specified our level-1 model by determining how time ought to be modeled. A linear fixed effect and random effect for time was most appropriate—a quadratic effect did not improve the model fit when compared via a −2 log likelihood (−2LL) deviance change test. Next, we added our sampling and COVID-19 covariates, which improved the model fit (*Χ*^2^ = 37.46, df = 3, *p* < .001). Finally, we added our predictors, which did significantly improve the model fit (*Χ*^2^ = 64.48, df = 8, *p* < .001). We also tested the further inclusion of interactions between our factors, but including these interactions did not substantially improve the model fit (*Χ*^2^ = 13.48, df = 6, *p* = .040) and no factors besides the intercept and change over time were significant in this interaction model; as such, we interpret the moderation model with only our predictors.

## 3. Results

### 3.1. Checks

No outliers were detected in the final dataset prior to analyses.

### 3.2. Direct Model: Belonging’s Change over Time

Contrary to our hypothesis, college students’ sense of belonging decreased over time from matriculation onward: b = −0.08, *SE* = 0.02, *t*(1455.16) = −3.89, *p* < .001, 95% CI: [−0.12, −0.04] (see Table 3). There was a significant random effect for time such that participants’ change in belonging over time varied significantly: τ_11_ = 0.01, *SE* = 0.002, Wald *Z* = 5.10, *p* < .001, 95% CI: [0.01, 0.02].

### 3.3. Direct Model: Covariate Effects

Potential covariates (study sampling variable or COVID indicator) did not play a role in students’ belonging (see Table 3). College students’ change in sense of belonging over time was not significantly affected by the COVID-19 pandemic: b = 0.02, *SE* = 0.02, *t*(1298.64) = 1.09, *p* = .274, 95% CI: [−0.02, 0.07] (see Table 3). The intercept was not affected either: b = −0.72, *SE* = 0.48, *t*(1231.03) = −1.52, *p* = .129, 95% CI: [−1.66, 0.21] (see Table 3). We also did not find an effect on the intercept for our risk factor associated with prescription drug misuse: b = 0.08, *SE* = 0.45, *t*(351.61) = 0.16, *p* = .871, 95% CI: [−0.83, 0.98] (see Table 3).

### 3.4. Moderation Model

Table 3 contains information about all parameters in the moderation model.

Greek affiliation showed no significant moderation: b = 0.02, *SE* = 0.02, *t*(344.84) = 0.77, *p* = .442, 95% CI: [−0.02, 0.05].

Participant sex did not moderate change in belonging over time: b = 0.02, *SE* = 0.02, *t*(276.86) = 1.13, *p* = .259, 95% CI: [−0.02, 0.06].

Results indicated that race/ethnicity significantly moderated the slope of students’ belonging over time: b = −0.06, *SE* = 0.02, *t*(350.48) = −2.92, *p* = .004, 95% CI: [−0.10, −0.02]. According to simple slope tests, students from minoritized racial/ethnic groups (b = −0.11, *t*[350] = −4.29, *p* < .001) showed steeper declines in belonging over time compared to White, non-Hispanic, or Latino/a students (b = −0.06, *t*[350] = −3.10, *p* = .002) (see Figure 1).

First-generation student status did not moderate change in belonging over time: b = 0.01, *SE* = 0.03, *t*(345.32) = 0.55, *p* = .583, 95% CI: [−0.04, 0.06].

## 4. Discussion

This study expands upon previous literature on college students’ health and development by testing how college students’ sense of belonging—a key predictor of improved student life—changes over students’ academic careers. We also examined how different factors affect changes to belonging. According to our findings—and contrary to our hypothesis—belonging decreased over our participants’ time in college. This is the single most important contribution we provide to the literature, which suggests that current students may see a drop in belonging over their college careers, contrary to educators’ naïve expectations (including our own). One demographic factor affected this change: race/ethnicity. Echoing findings showing that students from minoritized groups have challenges feeling a high sense of belonging on college campuses (e.g., [21,23,41]), our findings showed that our participants from minoritized racial/ethnic groups showed a steeper drop in their belonging than our White, non-Hispanic participants did. None of the other factors or covariates—including the COVID-19 pandemic’s onset—affected belonging’s change over time. These non-significant findings were surprising but demonstrate the strength of belonging’s downward trend among college students. So-called “belongingness boons”, like coming from a family of multigenerational college-goers or belonging to a Greek life organization, did not prove to be much help at all for maintaining belonging. On a positive note, we did not see significant differences between our self-identifying male and female participants, which may suggest a smaller gender gap in belonging than reported by other studies (e.g., [38]). Finally, the COVID-19 pandemic did not seem to affect our college students’ belonging levels in a significant way. This may suggest resilience among our sample that could be generalized to college students.

The general downward trend in belonging could be due to our choice of measures, given that we specifically asked participants about their sense of belonging in relation to the campus community. Many students, especially upper-level students, may find it more salient to identify with specific field or department communities rather than the larger campus community (e.g., upper-level engineering majors identify as engineers rather than general students). This identity salience transition could offer a social-identity-theory-informed reason for the drop in belonging [3]. This could also align with theories on social capital and community-building [46], which posit that networks form through norms of reciprocity and bonding: e.g., upper-level students are most likely bonding with students and faculty within their own departments rather than students and faculty of the campus at large.

Another possible explanation for the downward trend is that belonging naturally decreases as students habituate to the campus community, though this was not the case in a study by Ruedas-Gracia et al. [23]. Instead, they showed stability over three surveys assessed over 4 years among a smaller sample of private college students. One possible explanation is that there are important differences in how belonging changes among public vs. private college students, though differences in research design could also explain the disparity in findings. Our findings do align with the downward trend identified by Hausmann et al. [17] in the first year of Black and White college students’ career. As such, we might have discovered a general decrease in belonging that begins in the first year (as identified by Hausmann et al. [17]) then continues throughout the rest of the college career. Importantly, different factors may affect belonging’s decrease across the college career: for example, maybe the first year’s decrease caused by issues related to adjusting to college while the decrease in later years is due to social identity and social capital factors pushing students towards their department and away from the campus community.

Of note, Barringer et al. [16] showed a decrease in belonging across the COVID-19 pandemic’s onset among students from minoritized racial/ethnic groups in our same sample, though not from students overall. According to our study, college students’ belonging levels are highest shortly after matriculation—likely because of the belonging-enhancing activities associated with matriculation, like onboarding and welcome events. The largest drops in belonging appear between matriculation and later in the academic journey—something that would not have been picked up in the temporally constrained study by Barringer and colleagues [16]. The steeper drop in belonging associated with students from racial/ethnic minority groups in our study and the drop in belonging among the same during the course of the pandemic in Barringer et al. [16] remains worrisome though.

All of these factors point towards a theoretical framework that merges social identity and social capital theories along with applied interventions that universities use to increase belongingness. As identified in Hausmann et al. [17], students’ belonging levels dropped over their first year of college. This first stage of belonging’s decrease is likely driven by issues adjusting to the college environment: indeed, Hausmann et al. [17] showed a shallower drop in belonging among participants who received a belongingness intervention. Interventions to boost belongingness levels at the beginning of the collegiate experience are common on college campuses, being a primary focus for administrators and educators [17,32,33]. These belongingness interventions ought to raise belongingness levels among student participants higher than would be expected without these interventions, thus creating a place for a drop to occur as students habituate towards their “normal” level of belongingness after these interventions cease following the first and second semesters. After the first year, social capital and social identity factors begin to play a larger role than belongingness interventions. As shown by our study, belongingness to the campus community decreases over the remaining years of the collegiate experience, and this drop is steeper for students from minoritized racial and ethnic groups. From a social identity perspective [3], this could be due to a de-identification with the campus group as the student’s department, field, or research labs become more salient groups with which to identify. Social capital [46] further suggests that the groups with which we share norms will become more important and have more social capital to affect us. As students progress through college, we educators actively try to teach them norms associated with their particular fields, including writing norms, scholarship norms, and professional etiquette. These norms vary from discipline to discipline, so even if a college student were to remain in close contact with students in other disciplines, there will be a gulf of discipline-specific norms and social identities that will develop between them.

To illustrate this theoretical framework, we can imagine a prototypical psychology major entering their first year of college at a public four-year university. At first, our psychology major is thrown into a new world: commencement and “welcome week” events occur and introduce our psychology major to the campus community. The psychology major receives gifts of “swag” emblazoned with the university’s logo. If they live in student housing, then they may be paired with students in other majors whose only associative link is attendance at this university. All these factors ought to increase identification with the campus community at large and boost belonging [3]. But eventually, these activities cease, and the psychology major becomes adjusted to college life as they struggle with an increased academic workload and work to navigate their social circle. At this point, in our theoretical framework, the psychology major should be habituating towards a “normal” level of belongingness to the campus community that ought to remain relatively stable until they begin to advance through their major. Once the psychology major begins to participate in research labs, takes more major-specific classes, and obtains mentorship [26], then the salience [3] and social capital [46] of the campus community should be eclipsed by the salience and social capital of their department and major community. The psychology major will spend time with other psychology majors, with psychology faculty, and with the norms of psychological science. If you were to measure their belongingness to the campus community, the psychology major would show a two-phase drop, though if we measure belongingness in other ways, we would see another pattern emerge: belongingness to the campus community ought to *increase* following belongingness interventions at the beginning of the collegiate experience compared to their pre-college levels, then decrease as the student adjusts, then decrease further as the student swaps identification towards their discipline and major. Alternatively, belongingness towards their discipline and department community ought to increase counter to the decrease in belongingness towards the campus community. All of this should be exacerbated by any barriers to belonging that a student experiences.

Barriers to belonging for minoritized students have been identified in previous literature and include things like hostile racial/ethnic climate [28], perceptions of campus racial climate [31], loneliness, and marginality [20]. Students from salient minoritized groups would probably benefit from more targeted belongingness interventions to bolster their social identification and sense of belonging as a scholar in their fields or departments to combat the steeper drop in campus belonging (see [17]). Examples of some field-specific social identity bolstering interventions are reported by Kim and colleagues [18], Kuchynka and colleagues [47], Walton and Cohen [26], and Wilks [25]. Likewise, our findings highlight the importance of interventions to bolster upper-division students’ social identification levels. Some examples of interventions targeting upper-division students’ social identification include inclusive faculty mentorship [48], using pedagogies that teach useful academic skills in lower-level classes (see [41,49,50]), and creating an inclusive learning community in upper-level classes (see [7,51]).

### Strengths, Limitations, and Future Directions

Our study had many strengths, including a longitudinal design that allowed for parsing apart unique between- and within-person effects for belonging. Some limitations hampered the generalizability of our findings though, and it remains worthwhile for future researchers to consider them when designing studies further exploring belonging in higher education. In particular, the demographic makeup of our college campus did not allow us to examine the full breadth of diverse and intersecting identities U.S. college students hold [52,53]. The majority of our participants were White, non-Hispanic and/or Latino/a, and all participants identified according to cisgender identity (male or female). As eloquently identified by Barnett [54] and McIntocsh [55], the effects of background social identities within academia are oftentimes complex and intersecting when considering the privileges, powers, and positions afforded by things like gender, race, ethnicity, ability, sexuality, and age. We did account for a substantial portion of the between-person variance for belonging in our model, but there is still lots of variance left to account for. Future researchers should make efforts to collect information about important dimensions of identities that relate to belonging from participants, like ability and sexuality, in their measures and to collect sufficiently representative samples to examine effects within and between relevant social and cultural groups.

A related important limitation is that our findings were limited to a single university’s student body. As with other research in this area (e.g., [23]), it is hard to generalize findings from one university to other universities and colleges. Our university has unique characteristics that make it different from some universities and similar to others (see Section 2.1). In particular, our sample came from a university that is not designated as a minority-serving institution, meaning most of the student population is White. Also, the population is majority women, which may separate it from some other university settings. Finally, our sample was relatively young, reflecting prototypical college students but not all of the possible students that attend colleges. For future directions, we suggest that researchers and administrators make efforts to measure belonging among their student bodies so as to determine the direction of change in belonging within their particular context. This would aid educators and administrators to better target campaigns to increase belongingness among their student population and would help to identify students at particular risk for drops in belongingness, which might predict mental health issues [24] and academic problems [20]. Furthermore, this would directly allow for a meta-analysis of belonging across a wide range of universities and colleges, further informing considerations regarding the important topic of student belonging.

Another potential future direction is to more carefully examine how students from different disciplines experience changes in belongingness over time. Our sample was not collected with the intention of examining effects within and between specific academic disciplines. Comparisons between and within disciplines are important to understand the needs of specific students and to determine how targeted interventions affect those students. The interest in specific disciplines is well reflected by research on improving inclusivity within STEM fields [41,51,56] and would be supported in future research by obtaining information about students’ majors, departments, and career aspirations. This information could be analyzed via meta-analysis should educators and administrators heed our recommendations to collect and share these data with belonging researchers.

## 5. Conclusions

College students’ academic performance, health, and well-being are primary concerns for educators, administrators, and policymakers across all levels of influence. Our findings expand on previous literature about students’ sense of belonging by examining how this crucial construct changes over time and is associated with key factors. College students’ sense of belonging went down over time and racial/ethnic identities were associated with steeper drops for students from minoritized racial/ethnic groups. There was a random effect for change in belonging over time such that individuals showed significant variance in how much of a drop they experienced—this suggests that other factors may play a role in changes to students’ belongingness. We proposed a theoretical framework to explain this drop in belonging that incorporates research from social identity and social capital theories. Educators and researchers are encouraged to further research this topic to determine how their classrooms, departments, and institutions can help foster students’ belonging and maintain that belonging over time “so that the educational experience might be maximized for all students” ([57], p. 91).

## Figures and Tables

**Figure 1 ijerph-22-00472-f001:**
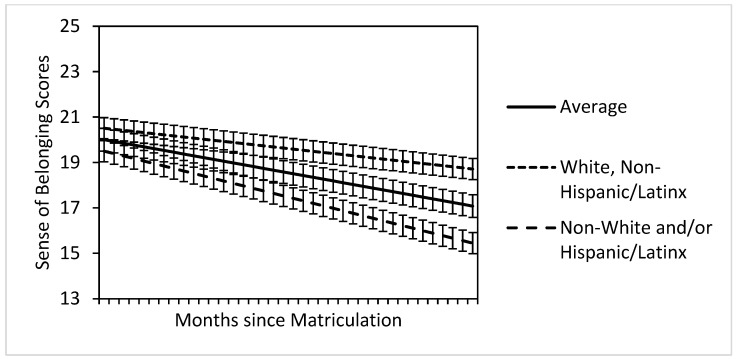
Change in belonging over time was moderated by race/ethnicity in the conditional model. Note. Error bars represent standard error. Race/ethnicity is effect-coded.

**Table 1 ijerph-22-00472-t001:** Participant demographics (*N* = 355).

Factor:	
Age at study baseline:	*M* = 19.48, *SD* = 0.70
Race:	
	White: 83.2%
	Asian: 5.7%
	Black: <2%
	American Indian or Alaskan Native: <2%
	Selected more than one race: 7.1%
	Selected “Other”: <2%
Ethnicity:	
	Hispanic or Latino/a background: 6.7%
	Not Hispanic or Latino/a background: 93.3%
Participant sex:	
	Female: 67.0%
	Male: 33.0%
First-generation status:	
	First-gen: 14.0%
	Non-first-gen: 85.4%
Greek-affiliation:	
	Greek-affiliation reported at least once: 34.8%
	No Greek-affiliation reported: 65.2%
Recent prescription drug misuse at study baseline:	
	Endorsed: 84.5%
	Not endorsed: 15.5%

**Table 2 ijerph-22-00472-t002:** Descriptive statistics for sense of belonging scores across waves T1–T5.

Wave	*N*	% Post-COVID Onset	*M*	*SD*	*α*
T1	355	0.0%	19.90	3.41	.84
T2	353	3.9%	19.68	4.04	.86
T3	348	28.4%	19.33	3.87	.87
T4	349	59.6%	19.03	3.98	.87
T5	346	81.5%	18.73	4.38	.87

Note. *M* = mean, *SD* = standard deviation, *α* = Cronbach’s alpha.

**Table 3 ijerph-22-00472-t003:** Fixed parameters from the direct and moderation models estimating college students’ sense of belonging over time.

Direct Model Estimating Belonging						
Fixed Effect	Estimate	*SE*	*t*	df	*p*	CI[LL, UL]
Intercept	20.78	0.49	42.25	1486.32	**<.001**	19.81, 21.74
Time	−0.08	0.02	−3.89	1455.16	**<.001**	−0.12, −0.04
Covariates						
Recent Rx misuse	0.08	0.45	0.16	351.61	.871	−0.83, 0.98
COVID indicator	−0.72	0.48	−1.52	1231.03	.129	−1.66, 0.21
Time*COVID indicator	0.02	0.02	1.09	1298.64	.274	−0.02, 0.07
Moderation Model Estimating Belonging						
Fixed Effect	Estimate	*SE*	*t*	df	*p*	CI[LL, UL]
Intercept	19.96	0.50	39.68	401.82	**<.001**	18.97, 20.95
Time	−0.07	0.02	−3.72	510.49	**<.001**	−0.11, −0.04
Covariates						
Recent Rx misuse	−0.59	0.46	−1.28	350.22	.200	−1.50, 0.32
COVID indicator	0.61	0.48	1.28	1238.50	.201	−0.33, 1.56
Time*COVID indicator	−0.02	0.02	−0.88	1307.87	.377	−0.06, 0.02
Level-1 Predictor and Moderation						
Greek-affiliation	0.43	0.35	1.26	531.41	.209	−0.24, 1.11
Time*Greek-affiliation	0.01	0.02	0.75	579.26	.455	−0.02, 0.05
Level-2 Predictors and Cross-Level Moderations						
Sex	0.32	0.39	0.81	348.28	.420	−0.45, 1.09
Time*Sex	−0.01	0.02	−0.29	348.74	.773	−0.04, 0.03
Race/ethnicity	−1.00	0.47	−2.15	347.66	**.032**	−1.92, −0.09
Time*race/ethnicity	−0.06	0.02	−2.92	350.48	**.004**	−0.10, −0.02
First-generation	−0.34	0.54	−0.63	345.17	.529	−1.40, 0.72
Time*First-Generation	0.004	0.03	0.16	355.05	.869	−0.05, 0.05

Note. *SE* = standard error; CI[LL, UL] = confidence interval [lower limit, upper limit]. * indicates an interaction effect. Bolded *p*-values indicated significant < .050. Recent Rx misuse is dummy-coded: 0 = not endorsed, 1 = endorsed. COVID indicator is dummy-coded: 0 = pre-COVID onset, 1 = post-COVID onset. Greek-affiliation is dummy-coded: 0 = not affiliated, 1 = affiliated. Sex is dummy-coded: 0 = male, 1 = female. Race/ethnicity is effect-coded: White, non-Hispanic and/or Latino/a = −0.5, racial/ethnic minority student = 0.5. First-generation is dummy-coded: 0 = not first-generation, 1 = first-generation status.

## Data Availability

The raw data, analysis code, and materials that support the findings of this study are available from the corresponding author upon reasonable request and application.

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
