# Peer review of "Evidence for College Students’ Decreasing Sense of Belonging over Time: Direct and Moderated Results"

_ijerph, 2025, doi:10.3390/ijerph22040472_

Round 1
Reviewer 1 Report
Comments and Suggestions for Authors
p.1.11 En dashes aren't em dashes. Please revise.
p.1.11/12 I find the mentioning of "recently" a bit odd. It has been quite some time that sense of belonging is considered crucial. I would suggest to rephrase the sentence so the time "stamp" (of the word "recently") goes away.
p.1.13 Place the N in italics if you are using APA 7 (see your Table 2; this has been implemented properly here).
p.18 "Findings expand..." = This sentence is not informative.
p.1.27/28 This statement requires a reference/ source.
p.1.44 The change in writing style is a bit odd (you go from passive to active). I would revise this and align it with the aforementioned (thus, write in a passive voice).
p.Table 1 is aligned a bit odd. Please check.
p.5.180 You do not have 1757 participants. Please revise. Your sample has been stated earlier in that paragraph.
If you are using APA 7, please revise your reference list according to those rules.
Author Response
Reviewer 1:
- 1.11 En dashes aren't em dashes. Please revise.
- We appreciate Reviewer 1’s careful attention to detail. We have revised en dashes to em dashes in the abstract on p. 1.11
- 1.11/12 I find the mentioning of "recently" a bit odd. It has been quite some time that sense of belonging is considered crucial. I would suggest to rephrase the sentence so the time "stamp" (of the word "recently") goes away.
- We agree with the reviewer’s assessment that student sense of belonging has received attention for a long while now, and that the time stamp of “recently” incorrectly suggests otherwise. As such, we have removed the phrase “recently received attention” on p. 1.12 so as to focus on the accurate fact that change in student belonging needs further examination.
- 1.13 Place the N in italics if you are using APA 7 (see your Table 2; this has been implemented properly here).
- Once again, we appreciate Reviewer 1’s careful attention to detail. We have italicized N in this place and in all other places throughout the text where it appears.
- 18 "Findings expand..." = This sentence is not informative.
- We agree that this sentence did not add much to the abstract and have struck it entirely.
- 1.27/28 This statement requires a reference/ source.
- We appreciate Reviewer 1’s focus on improving the strength of our introduction. While we have cited 6 references in this sentence to support this statement already, we do believe that further references will improve support for the statement that “college students are an especially important student population”. As such, we have added a new citation on p. 1.28. The new reference [1] is as follows:
- Lederer, A.M., & Oswalt, S.B. (2017). The value of college health promotion: A critical population and setting for improving the public’s health. American Journal of Health Education, 48, 215-218. https://doi.org/10.1080/19325037.2017.1316692
- 1.44 The change in writing style is a bit odd (you go from passive to active). I would revise this and align it with the aforementioned (thus, write in a passive voice).
- We have modified the text according to this recommendation.
- Table 1 is aligned a bit odd. Please check.
- We appreciate Reviewer 1’s keen eye for formatting. We modified the alignment for Table 1, along with all of the other tables and figures in the text.
- 5.180 You do not have 1757 participants. Please revise. Your sample has been stated earlier in that paragraph.
- We appreciate Reviewer 1’s suggestion regarding our statement about sample size. We agree that we did not have 1757 participants, so we have removed the N from this statement to make it clear that we had 1757 total data points.
- If you are using APA 7, please revise your reference list according to those rules.
- Once again, we appreciate Reviewer 1’s keen eye for formatting. We have combed through our reference list carefully to align it with both MDPI’s and the American Psychological Association’s formatting guidelines. For example, we have added doi’s where possible, revised some formatting, and fixed capitalization errors.
Reviewer 2 Report
Comments and Suggestions for Authors
This is a well-done study on the sense of belonging of college students referring to the social ties with the 'alma mater' institution. Data collection is longitudinal, which presents a major strength of the manuscript. Methods are rather simplistic (simple moderation), but are appropriate for the objective of the study.
However, I have some questions and comments. First, there is a principal issue of including the drug abuse into your research design. It has been mentioned in the funding section that your study has been funded by NIH, under a project, dealing with drug abuse. Therefore, I am aware that you needed to include this variable due to your funding requirements. At the other hand, I am concerned about the ethical aspect of the study, as the drug abuse does not seem to have any major role in this study, in the theoretical sense. It is just a covariate, any other socio-economic, or behavioral variable, might do in the same context. You need to declare any potential ethical conflicts and make sure that your research design serves the purpose of the grant. This is not an issue that can be dealt with implicitly, within a single footnote, especially concerning that participants have received quite generous amounts of remuneration for participating in the study. I feel that all those circumstances should be well explained to make sure the ethical background is sorted out before publishing.
Otherwise, there are some theoretical and formal limitations of the manuscript. First, it is nice to see that your empirical results are different that the theory posits, as this could open up some theorizing opportunities. However, those have not been used at all - in the theoretical background, you do not even mention arguments which could support your empirical results. There are some theoretical points concerning the diminishing sense of belonging in the discussion, but these do not frame your findings.
I am unsure about your background, but some sociological theories concerning social capital, sense of belonging, and theory of community-building could be used here.
You should also mention the limitation of data collection at a single university. I also miss the practical implications of the study.
From the formal viewpoint, you need to follow the MDPI style formatting more closely.
Author Response
- This is a well-done study on the sense of belonging of college students referring to the social ties with the 'alma mater' institution. Data collection is longitudinal, which presents a major strength of the manuscript. Methods are rather simplistic (simple moderation), but are appropriate for the objective of the study.
- We appreciate Reviewer 2’s assessment of our study.
- However, I have some questions and comments. First, there is a principal issue of including the drug abuse into your research design. It has been mentioned in the funding section that your study has been funded by NIH, under a project, dealing with drug abuse. Therefore, I am aware that you needed to include this variable due to your funding requirements. At the other hand, I am concerned about the ethical aspect of the study, as the drug abuse does not seem to have any major role in this study, in the theoretical sense. It is just a covariate, any other socio-economic, or behavioral variable, might do in the same context. You need to declare any potential ethical conflicts and make sure that your research design serves the purpose of the grant. This is not an issue that can be dealt with implicitly, within a single footnote, especially concerning that participants have received quite generous amounts of remuneration for participating in the study. I feel that all those circumstances should be well explained to make sure the ethical background is sorted out before publishing.
- We appreciate Reviewer 2’s concern about our sample and potential ramifications with our funding agency. We believe it is important to mention at this point that the funding agency did not require us to include the covariate of recent prescription drug misuse in our current study. Instead, we thought that including this covariate strengthened the research design and made our research design more transparent. Furthermore, this study is a secondary analysis which takes advantage of a larger study that was done for a different purpose. It is a common strategy with large NIH-funded studies to perform these sorts of secondary analyses. As such, we thought it best to control for the larger study’s sampling strategy by including the study enrollment variable as a covariate. We have emphasized this matter in a new statement found on p. 6.210-213. Finally, regarding participant payment, our compensation approach was in line with other funded longitudinal studies (at the time of the study design) and approved by our IRB.
- If our understanding of Reviewer 2’s comment is wrong, then we would appreciate further clarification so that we can address it more effectively.
- Otherwise, there are some theoretical and formal limitations of the manuscript. First, it is nice to see that your empirical results are different that the theory posits, as this could open up some theorizing opportunities. However, those have not been used at all - in the theoretical background, you do not even mention arguments which could support your empirical results. There are some theoretical points concerning the diminishing sense of belonging in the discussion, but these do not frame your findings.
- We agree that our findings were not strongly supported in our literature review or theoretical background. As such, we have taken efforts to emphasize previous literature that shows a decrease in belonging more strongly in this section. For example, on p. 2.80-82 we have added the following statement after presenting findings that showed a decrease in belonging during the first year of college (Hausmann et al., 2017): “This decline in belonging may progress beyond the first year of college. If so, then we would expect to see belonging decrease over students’ college careers in a longer longitudinal investigation.” Furthermore, we added a new statement connecting our findings to Hausmann et al. (2017) in our discussion on p. 10.357-364.
- We also more clearly aligned our hypothesis about belonging remaining stable with the primary theoretical contributor: Ruedas-Garcia et al. (2023).
- I am unsure about your background, but some sociological theories concerning social capital, sense of belonging, and theory of community-building could be used here.
- We appreciate Reviewer 2’s suggestion of this literature. We have added a statement and reference to social capital and community building in our discussion on p. 10.346-350.
- You should also mention the limitation of data collection at a single university. I also miss the practical implications of the study.
- We appreciate the attention to this particular limitation, and we have added it to our list of limitations on p. 11.406-414. We also added a statement to this new paragraph that emphasizes the practical implications of research in this area in general and our study in particular.
- From the formal viewpoint, you need to follow the MDPI style formatting more closely.
- We appreciate Reviewer 2’s keen eye for formatting. We have attempted to bring our document into line with MDPI’s proper style formatting. For example, we have adjusted the tables and figures to make them fit within the proper margins.
Round 2
Reviewer 2 Report
Comments and Suggestions for Authors
Dear authors,
while I appreciate the progress made with the current revision of your manuscript, there are, still, several points which need clarification:
- While you have included some references concerning the declining sense of belonging, it would be appreciated if you introduced a more robust theoretical framework, trying to explain your findings. There could be a range of factors, contributing to such research results (such as lack of institutional support, shifting identity focus, systemic inequities, etc.).
- I appreciate your comment in the formal reviewer replies ("this study is a secondary analysis which takes advantage of a larger study that was done for a different purpose. It is a common strategy with large NIH-funded studies to perform these sorts of secondary analyses. As such, we thought it best to control for the larger study’s sampling strategy by including the study enrollment variable as a covariate"). However, this is not entirely clear from the revised text of your manuscript. Please indicate the prior research practice of secondary analysis with appropriate citations. You should also justify why you prioritize the drug misuse over other covariates and what is your theoretical support.
- You have not explicitly acknowledged the limitation of high monetary compensations of study participants and their potential influence to overemphasizing some specific socio-economic, or other participant background in you sample, or to potentially skewing some research results. Please discuss.
- Limitations of data collection at a single university are, also, understated, and should be discussed more extensively.
- Formulate more actionable practical limitations.
- Please make sure that your formatting strictly adheres to MDPI technical requirements.
Author Response
- While you have included some references concerning the declining sense of belonging, it would be appreciated if you introduced a more robust theoretical framework, trying to explain your findings. There could be a range of factors, contributing to such research results (such as lack of institutional support, shifting identity focus, systemic inequities, etc.).
- We appreciate the reviewer’s suggestion to explicate our theoretical framework more completely. We have taken the opportunity to do so and added two paragraphs spanning pp. 12-13 lines 386-436. In particular, we merge social identity and social capital theories along with applied interventions that universities use to increase belongingness.
- For ease of access, those paragraphs are also pasted below:
- “All of these factors point towards a theoretical framework that merges social identity and social capital theories along with applied interventions that universities use to in-crease belongingness. As identified in Hausmann et al. [17], students’ belonging levels dropped over their first year of college. This first stage of belonging’s decrease is likely driven by issues adjusting to the college environment – indeed, Hausmann et al. [17] showed a shallower drop in belonging among participants who received a belongingness intervention. Interventions to boost belongingness levels at the beginning of the collegiate experience are common on college campuses, being a primary focus for administrators and educators [17, 32, 33]. These belongingness interventions ought to raise belongingness levels among student participants higher than would be expected without these interventions, thus creating a place for a drop to occur as students habituate towards their “nor-mal” level of belongingness after these interventions cease following the first and second semesters. After the first year, social capital and social identity factors begin to play a larger role than belongingness interventions. As shown by our study, belongingness to the campus community decreases over the remaining years of the collegiate experience, and this drop is steeper for students from minoritized racial and ethnic groups. From a social identity perspective [3], this could be due to a de-identification with the campus group as the student’s department, field, or research labs become more salient groups with which to identify. Social capital [54] further suggests that the groups with which we share norms will become more important and have more social capital to affect us. As students progress through college, we educators actively try to teach them norms associated with their particular fields, including writing norms, scholarship norms, and professional etiquette. These norms vary from discipline to discipline, so even if a college student were to remain in close contact with students in other disciplines, there will be a gulf of discipline-specific norms and social identities that will develop between them.
- “To illustrate this theoretical framework, we can imagine a prototypical psychology major entering their first year of college at a public four-year university. At first, our psychology major is thrown into a new world: commencement and “welcome week” events occur and introduce our psychology major to the campus community. The psychology major receives gifts of “swag” emblazoned with the university’s logo. If they live in student housing, then they may be paired with students in other majors whose only associative link is attendance at this university. All these factors ought to increase identification with the campus community at large and boost belonging [3]. But eventually, these activities cease, and the psychology major becomes adjusted to college life as they struggle through an increased academic workload and work to navigate their social circle. At this point, in our theoretical framework, the psychology major should be habituating towards a “normal” level of belongingness to the campus community that ought to remain relatively stable until they begin to advance through their major. Once the psychology major begins to participate in research labs, takes more major-specific classes, and obtains mentorship [26], then the salience [3] and social capital [54] of the campus community should be eclipsed by the salience and social capital of their department and major community. The psychology major will spend time with other psychology majors, with psychology faculty, and with the norms of psychological science. If you were to measure their belongingness to the campus community, the psychology major would show a two-phase drop, though if we measure belongingness in other ways we would see another pattern emerge: belongingness to the campus community ought to increase following belongingness interventions at the beginning of the collegiate experience compared to their pre-college levels, then decrease as the student adjusts, then decrease further as the student swaps identification towards their discipline and major. Alternatively, belongingness towards their discipline and department community ought to increase in a counter to the decrease of belongingness towards the campus community. All of this should be exacerbated by any barriers to belonging that a student experiences.”
- I appreciate your comment in the formal reviewer replies ("this study is a secondary analysis which takes advantage of a larger study that was done for a different purpose. It is a common strategy with large NIH-funded studies to perform these sorts of secondary analyses. As such, we thought it best to control for the larger study’s sampling strategy by including the study enrollment variable as a covariate"). However, this is not entirely clear from the revised text of your manuscript. Please indicate the prior research practice of secondary analysis with appropriate citations. You should also justify why you prioritize the drug misuse over other covariates and what is your theoretical support.
- We appreciate Reviewer 2’s concern over potential confusion for the reader regarding secondary analysis. We have added a new paragraph to the 1.3 Current Study subsection (p. 4 lines 144-147) highlighting how we are conducting a secondary analysis and mentioning literature that references secondary analysis, including two previous secondary analysis studies using this same dataset with the same covariate (Barringer et al., 2023; Brown et al., 2024) along with an overview of NIH primary and secondary analyses conducted between 2012-2019 (Murray et al., 2021). Finaly, we provided citation to an article which references best practices for conducting secondary analyses (Cheng & Phillips, 2014).
- You have not explicitly acknowledged the limitation of high monetary compensations of study participants and their potential influence to overemphasizing some specific socio-economic, or other participant background in you sample, or to potentially skewing some research results. Please discuss.
- We appreciate Reviewer 2’s concern about monetary compensation, though we counterpoint that our compensation ought not to be considered unethically high for the original study’s conception. Indeed, our compensation strategy was approved by our Institutional Review Board prior to study recruitment. Furthermore, our payment aligns with current ethical considerations of social beneficence, as outlined by RóżyÅ„ska (2022). We have added statements addressing this into our text directly when we present our compensation strategy on p. 5 lines 186-188.
- Limitations of data collection at a single university are, also, understated, and should be discussed more extensively.
- We appreciate Reviewer 2’s concern about this particular limitation. We have added further clarification highlighting how limiting this limitation is given our particular sample (p. 13 lines 469-475). We further emphasize our recommended future direction of more research in this area within more institutions.
- Formulate more actionable practical limitations.
- While we agree that limitations ought to lead to practical future directions, we are not sure if this is what Reviewer 2 is recommending in this comment. If not, we would appreciate further clarification. Assuming that is the intent of this comment, we have added direct recommendations to subsection 4.1 Strengths, Limitations, and Future Directions. Our most actionable recommendations are to measure more dimensions of identity in surveys (p. 13 lines 464-465), measure information regarding students’ majors and career aspirations (p. 14 lines 490-494), and to collect these data across a wide range of institutions to allow for meta-analysis of students’ belonging (pp. 13-14 lines 481-484).
- Please make sure that your formatting strictly adheres to MDPI technical requirements.
- We appreciate Reviewer 2’s attention to detail and have made every effort to align our formatting strictly to MDPI’s technical requirements. In addition to using the Microsoft Word template provided by MDPI, we have carefully gone through the layout guide (https://www.mdpi.com/authors/layout). Our title is in proper format, our author names are in proper format, and our abstract is in proper format. We have followed the IMRAD structure for the overall structure of the paper. MDPI has no specific recommendations for paragraph structure, but we have endeavored to use the best grammar and typical conventions for English writing. We use three levels of headings, as per the recommendations for headings and sections, and have them properly in title case. We have checked through the tense used in each section and have made changes in line with MDPI’s recommendations for tenses. We have changed the spacing for our statistical results and the note in Table 3. All other matters appear to match formatting.